# Brain Hemispheric Asymmetry in Schizophrenia and Bipolar Disorder

**DOI:** 10.3390/jcm12103421

**Published:** 2023-05-12

**Authors:** Diogo Pinto, Ricardo Martins, António Macedo, Miguel Castelo Branco, João Valente Duarte, Nuno Madeira

**Affiliations:** 1Faculty of Medicine, University of Coimbra (UC), 3004-504 Coimbra, Portugal; diogopinto@outlook.pt (D.P.); ricardo.martins@uc.pt (R.M.); joao.v.duarte@fmed.uc.pt (J.V.D.); 2Coimbra Institute for Biomedical Imaging and Translational Research (CIBIT), Institute of Nuclear Sciences Applied to Health (ICNAS), University of Coimbra, 3000-548 Coimbra, Portugal; 3Department of Psychiatry, Centro Hospitalar e Universitário de Coimbra (CHUC), 3000-075 Coimbra, Portugal

**Keywords:** schizophrenia, bipolar disorder, hemispheric asymmetry, magnetic resonance imaging

## Abstract

Background: This study aimed to compare brain asymmetry in patients with schizophrenia (SCZ), bipolar disorder (BPD), and healthy controls to test whether asymmetry patterns could discriminate and set boundaries between two partially overlapping severe mental disorders. Methods: We applied a fully automated voxel-based morphometry (VBM) approach to assess structural brain hemispheric asymmetry in magnetic resonance imaging (MRI) anatomical scans in 60 participants (SCZ = 20; BP = 20; healthy controls = 20), all right-handed and matched for gender, age, and education. Results: Significant differences in gray matter asymmetry were found between patients with SCZ and BPD, between SCZ patients and healthy controls (HC), and between BPD patients and HC. We found a higher asymmetry index (AI) in BPD patients when compared to SCZ in Brodmann areas 6, 11, and 37 and anterior cingulate cortex and an AI higher in SCZ patients when compared to BPD in the cerebellum. Conclusion: Our study found significant differences in brain asymmetry between patients with SCZ and BPD. These promising results could be translated to clinical practice, given that structural brain changes detected by MRI are good candidates for exploration as biological markers for differential diagnosis, besides helping to understand disease-specific abnormalities.

## 1. Introduction

Schizophrenia (SCZ) is characterized by positive psychotic symptoms (hallucinations, delusions, formal thought disorder), negative symptoms (flat affect, poor motivation, avolition), and, in many cases, social and occupational decline that can lead to life-changing consequences even in those who have good outcomes [1,2].

Bipolar disorder (BPD) is a mood disorder commonly characterized by nonspecific symptoms, mood lability, or a depressive episode. BPD can be defined by episodes of mania (BPD type 1) or characterized by at least one hypomanic episode and one major depressive episode (BPD type 2). Common symptoms during manic episodes are grandiosity, hyperactivity, psychotic symptoms, and expansive mood, whereas, through depressive episodes, they are sadness, decreased energy, and social withdrawal [3].

Furthermore, BPD can be easily confused with SCZ, particularly when psychotic symptoms are prominent due to overlap in clinical presentation [4,5,6]. In addition, treatment options are highly different for both pathologies and might impact mortality and morbidity, so it is crucial to define biomarkers, especially from the earliest stages of the disease, that can help differentiate each pathology [4]. Since several studies have described a high genetic overlap between SCZ and BPD [4,5,7], and specific biomarkers for BPD are not yet available [3], structural brain changes detected by magnetic resonance imaging (MRI) are good candidates for exploration as biological markers to distinguish both conditions [7].

Left–right hemispheric asymmetry is an important aspect of human brain organization for diverse cognitive functions such as language, social cognition, and executive and affective processes. Hemispheric asymmetry can be altered in various aspects, such as anatomical gray and white matter measures, functional and structural connectivity, behavioral associations, and gyrification [8]. Whereas some studies found that differences in SCZ are more evident in the left hemisphere, while changes in BPD are more significant in the right hemisphere [7], others have provided evidence of common abnormalities in brain structures in patients with SCZ and BPD [5].

Global gray matter (GM) asymmetry is related to developmental stability and is a valuable indicator of disturbance during neurodevelopment. In addition, global GM asymmetry was associated with avolition and anxiety. However, other symptoms, such as hallucinations, were not [9]. Structural brain modifications and disrupted interregional connections have been related to many characteristic symptoms and cognitive impairment demonstrated in patients with SCZ [10].

It is widely described by evidence that there is atypical lateralization in brain structure, cortical thickness, and brain volumetric asymmetries in SCZ and BPD either post-mortem or in neuroimaging investigations, suggesting that both pathologies may be the result of a failure of the normal lateralization process of the brain. However, GM volume reductions are consistently more extensive in SCZ, except in BPD patients with comorbid psychosis [4,5,9,11,12]. Diverse studies reported that patients with either pathology have more global GM asymmetry than healthy controls (HC) with reduced cortical volume and thickness. Cortical volume variations in SCZ are explained by changes in cortical thickness and surface area, while in BPD, they are mostly justified by cortical thinning [4,9,13,14,15]. It is also important to note that no differences were detected in cortex thickness and surface area between patients with BPD type 1 and type 2 [15]. It is important to consider that cortical thickness in SCZ in prefrontal and temporal cortical areas is modulated by antipsychotic (AP) treatment and illness duration [4].

Brain changes with extensive GM loss in SCZ patients when compared to BPD patients include neocortical structures (prefrontal and temporal cortices), limbic (amygdala, hippocampus, thalamus), paralimbic (anterior cingulate, insula), and, to some degree, the cerebellum and increased GM volume in basal ganglia (globus pallidus) in SCZ but not in BPD, while changes in BPD are more restricted to paralimbic regions, frontal cortex, and thalamus, areas involved in reward and pain systems and emotional regulation [1,4,7,11,12,13,14,16,17,18,19]. Despite SCZ presenting GM volume deficits in multiple cortical and subcortical areas [10], patients with SCZ, compared with HC, show more symmetry in subcortical areas such as the caudate nucleus and hippocampus, except that the thalamus and HC seem to be more symmetric in cortical areas such as prefrontal region and insula [9].

Enumerating other changes that could contribute to hemispheric brain asymmetry, although anatomical brain changes affect GM more than white matter (WM), WM connectivity dysfunctions are also present in SCZ and BPD [5] in alignment with GM variations and influence connectivity in prefrontal and limbic regions in SCZ [11].

Regarding the gyrification index, studies describe divergent gyrification of the left supramarginal gyrus with increased gyrification in BPD, reduced gyrification in SCZ, and decreased gyrification of the right inferior frontal gyrus in SCZ [4].

Taking all previous studies and knowledge into consideration and exploring the lack of studies directly comparing hemispheric asymmetry in SCZ and BPD, our main purpose was to directly compare carefully defined matched groups of SCZ vs. BPD patients and HC to find possible structural asymmetry-based biomarkers that can help differential diagnosis between pathologies and guide future research.

## 2. Materials and Methods

### 2.1. Participants

This study aimed to compare brain magnetic resonance imaging scans of individuals with SCZ (*n* = 20), BPD (*n* = 20), and HC (*n* = 20), all right-handed and matched for age (18–54), gender, and education. Participants were recruited from a major university hospital, and inclusion criteria for clinical groups were (a) ICD-10 criteria for SCZ or BPD confirmed through a direct interview by an experienced psychiatrist and reviewing medical records; (b) capacity to consent; (c) age between 18 and 54; (d) right-handedness through evaluation with the Edinburgh Handedness Inventory [20]; (e) clinical stability in the last 3 months prior to enrolment with unchanged medication for a similar period. Exclusion criteria were (a) neurological or medical comorbidity (e.g., head trauma, epilepsy, neurodevelopmental disorders); (b) MRI contra-indications; (c) substance abuse/dependence. Patients’ clinical assessment included the following instruments: the Schizo-Bipolar Scale [21], developed to capture the dimensional interaction between affective symptoms and psychosis; the Insight and Treatment Attitudes Questionnaire (ITAQ), to assess insight [22]; the Personal and Social Performance Scale (PSP) [23], addressing functioning; and the Brief Psychiatric Rating Scale (BPRS) [24], for general psychopathology assessment. Current AP exposure in SCZ and BPD patients was calculated through chlorpromazine equivalents (CPZE) [25]. Control individuals matched for age, gender, and education were recruited from the institution’s workers and their relatives with a brief interview excluding personal or first-degree family history of mental disorders, namely SCZ or BPD, in addition to general exclusion criteria. The study was approved by the local Ethics Committee of the Faculty of Medicine of the University of Coimbra (ref. CE-010/2014), conducted in accordance with the Declaration of Helsinki. All participants provided written informed consent.

### 2.2. MRI Acquisition

Magnetic resonance imaging data were collected with a Siemens Magnetom TIM Trio 3T scanner (Siemens, Munich, Germany) with a phased array 12-channel birdcage head coil. The magnetic resonance imaging dataset consisted of a 3D anatomical T1-weighted MPRAGE (magnetization-prepared rapid gradient echo) pulse sequence (TR 2530 ms; TE 3.42 ms; TI 1100 ms; flip angle 7°; 176 single-shot interleaved slices [no inter-slice gap] with isotropic voxel size 1 × 1 × 1 mm; FOV 256 mm) of all 60 participants.

### 2.3. Processing and Asymmetry Index

The data were processed and analyzed using a recently published protocol—“A 12-step user guide for analyzing voxel-wise gray matter asymmetries in statistical parametric mapping”— to find biomarkers useful in differential diagnosis [26]. This protocol describes a fully automated voxel-based morphometry (VBM) approach to assess structural brain hemispheric asymmetry, which is capable of capturing GM asymmetries with extremely high regional specificity [26,27].

All images were processed and analyzed using SPM8 and VBM8 toolbox. We performed tissue segmentation into three voxel classes (GM, WM, and cerebrospinal fluid), and we flipped tissue segments at midline. We also performed spatial normalization using the DARTEL (Diffeomorphic Anatomical Registration Through Exponentiated Lie Algebra) approach. DARTEL is a high-dimensional normalization algorithm, provided by SPM8, that has been shown to secure a better registration across brains than the Statistical Parametric Mapping (SPM) default normalization [28], because asymmetry VBM needs an accurate voxel-wise correspondence not only between brains, used in standard VBM, but also across hemispheres. Then, we warped the original and flipped tissue segments to the symmetric DARTEL template. Combined with a right hemisphere mask, we calculated the AI images and discarded the left hemisphere. To control the possible impact of noise, spatial smoothing was also implemented with a size of the smoothing kernel of 8 mm full width at half maximum (FWHM). Spatial smoothing certifies that random errors have a Gaussian distribution (a precondition for parametric tests). Processing also included quality assurance since all images were visually inspected before and during pre-processing to guarantee that no parts of the brain were cut off, wrapped, or distorted, and that images had no artifacts. Finally, the asymmetry index (AI) was quantified in each voxel by comparing original and flipped GM segments using the following equation:(1)$$Asymmetry\;Index=\frac{\left(original-flipped\right)}{0.5\times (original+flipped)}$$

### 2.4. Statistical Analysis

We performed statistical analyses in SPM8 statistical module by applying a two-sample Student’s *t*-test to assess differences in GM asymmetry between the group of patients with SCZ and the group with BPD and between each group of patients individually and HC. The spatially normalized and smoothed GM segments constitute the input for the voxel-wise statistical analyses [29]. We tested group differences and considered differences significant at a voxel-level threshold of *p* < 0.001. A positive AI indicates rightward asymmetry, whereas a negative AI indicates leftward asymmetry, with higher absolute values reflecting stronger asymmetry [26]. An ANCOVA between SCZ and BPD patients’ left and right gray matter volumes in significant clusters with total intracranial volume as a covariate was also performed to find whether the volume differences in one hemisphere are significatively different between SCZ and BPD patients.

Clusters with significant differences in GM asymmetry index were labelled according to their MNI coordinates and the corresponding anatomical area of the brain using the software package GingerALE (version 3.0.2; http://www.brainmap.org/ale/, accessed on 1 January 2023).

## 3. Results

### 3.1. Demographic and Clinical Characterization

Clinical and demographic data of the study participants are summarized in Table 1. All patients with SCZ were on AP medication, mostly atypical APs: first-generation AP (*n* = 2), one second-generation AP (*n* = 16), and a combination of two second-generation AP (*n* = 2). Among the patients with BPD, most (*n* = 18) were on regular mood stabilizing (MD) medication: MD monotherapy (*n* = 7), MD in association (*n* = 1), MD and atypical AP (*n* = 4), monotherapy with atypical AP (*n* = 4), and a combination of atypical AP (*n* = 2). One patient was medicated with lithium, and two were stable without medication.

Groups were balanced for age and gender, with exactly the same within-group distribution (χ^2^ = 0.000, *p* = 1.000). SCZ and BPD groups had no relevant clinical or demographic differences besides AP exposure, which was higher in SCZ patients (*p* = 0.032). Concerning psychopathological evaluation, SCZ patients had superior (*p* < 0.001) general psychopathologic scores and worse (*p* = 0.001) functioning than BPD patients. Individuals in both clinical groups were either in remission or sustained clinical stability, as shown by mean BPRS scores for patients with SCZ (35.65 ± 6.41) and BPD (29.11 ± 2.61). Differences between the two groups were statistically significant (*p* < 0.001). SCZ patients had lower social functioning than BPD patients using PSP scores (*p* = 0.001). BPD group had higher (*p* = 0.044) insight than SCZ patients. The SCZ group had much higher scores (*p* < 0.001) on the Schizo-Bipolar Scale than the BPD group as expected, because higher scores are associated with prototypical SCZ syndromes, whereas paradigmatic BPD patients score lower [4].

### 3.2. Voxel-Based Morphometry—Asymmetry Index SCZ vs. BPD

The contrast of higher GM asymmetry index in SCZ compared with BPD, performed at the whole brain level, revealed a significant group difference (*p* < 0.001) in one cluster with a cluster size of 82 voxels, at the MNI coordinates (11, −34, −26) and (6, −45, −30), corresponding to right cerebellar hemisphere, anterior lobe, gyrus culmen. The analysis of the cluster-specific mean AI revealed a stronger rightward asymmetry in the SCZ group than in the BPD group. The analysis between SCZ and BPD groups of this cluster’s specific GM volume showed that, although not statistically significant (*p* = 0.080), there is a trend for the SCZ group to have lower GM volume in the left hemisphere. No statistically significant differences were found in the right hemisphere (*p* = 0.406). These results are described in Figure 1, Table 2 and Table 3.

The contrast of higher GM asymmetry index in BPD compared to SCZ, performed at the whole brain level, revealed significant group differences (*p* < 0.001) in four clusters described in Table 2 and Figure 2. Evaluating each cluster’s specific mean AI revealed a stronger rightward asymmetry in the BPD group in all clusters. The analysis between the SCZ and BPD groups’ GM volume showed that the SCZ group had statistically significantly higher GM volume in the left hemisphere in Brodmann area 6 (BA6) (*p* = 0.012) and Brodmann area 11 (BA11) and anterior cingulate cortex (ACC) (*p* = 0.018), while the BPD group had statistically significant, or a tendency to, higher GM volume in the right hemisphere in BA6 (*p* = 0.007 and *p* = 0.078), in Brodmann area 37 (BA37) (*p* = 0.002), and in BA11 and ACC (*p* = 0.044). These GM volume differences are described in Table 3.

### 3.3. Voxel-Based Morphometry—Asymmetry Index SCZ vs. HC and BPD vs. HC

We performed, at a whole brain level, two-sample Student’s *t*-tests of the hypotheses SCZ > HC, SCZ < HC, BPD > HC, and BPD < HC. Table 2 describes all clusters showing significant group differences in voxel-wise GM asymmetry.

## 4. Discussion

To our knowledge, this study is the first to employ a standard protocol [26] to directly investigate differences in GM volume asymmetry in patients with SCZ and BPD in the same study. Our study found significant differences in GM asymmetry between patients with SCZ and BPD, as well as significant differences between SCZ patients and HC and between BPD patients and HC. Notably, the clusters with significant AI differences between each patient group and HC are different from the ones revealed when directly comparing SCZ and BPD, thus conferring discriminative specificity to these results and highlighting the potential of this measure of GM volume asymmetry to constitute a brain marker that might help in the differential diagnosis between these two archetypal psychiatric diseases.

The study found that the cerebellum of patients with SCZ showed a higher asymmetry index (AI) compared to BPD patients, indicating stronger rightward asymmetry in SCZ patients. The SCZ group also had lower left hemisphere volume, specifically in the gyrus culmen in the anterior lobe of the right cerebellar hemisphere, compared to BPD.

Although the evidence for cerebellar abnormalities in SCZ is getting stronger, it is less extensive than for other brain regions, such as frontal and temporal cortices [17]. However, in BPD, few studies demonstrate differences in the cerebellum. Only recently, implicit motor sequence learning impairment, already shown in SCZ, was connected to cerebellar abnormalities in BPD [30].

The cerebellum is not only involved in motor coordination; it relates to many cerebral cortical regions and plays a role in various cognitive functions such as facial recognition, emotion attribution, attention, verbal and motor learning, sensory discrimination, problem-solving, memory, and visual perception [2,17,31,32,33,34].

Cerebellar deficiency and a disturbed pre-frontal-thalamic-cerebellar circuit could lead to cortical malfunction and contribute to the diversity of symptoms and cognitive dysfunctions observed in SCZ [2,17,35]. Structural deficits in the cerebellum and related networks are associated with abnormal posture, proprioception, eyeblink conditioning, vestibular ocular reflex, neurological soft signs, and negative symptoms in SCZ [30,31,35,36,37].

Purkinje cells, which integrate information on the self and external influences, are decreased in size or density in SCZ, which may cause various symptoms such as hallucinations, thought insertion, and replaced control of will [17,31]. Our findings of left hemisphere GM volume decrease in SCZ are in accordance with this literature. However, further studies are needed to understand the impact of this finding on the differential diagnosis and the function of the cerebellum in BPD’s pathophysiology.

Regarding the hypothesis of higher GM volume asymmetry in BPD compared to SCZ (contrast SCZ < BPD), several clusters were found to have significant differences. On the one hand, differences were found in two clusters corresponding to BA6 in the medial frontal gyrus of the right hemisphere.

BA6 is composed of a lateral portion—premotor cortex, responsible for planning movements, correction of postural adjustments, and locomotion and social cognition skills as part of the mirror neuron system [6,38]—and a medial portion—supplementary motor cortex, responsible for planning, initiation, and anticipation of body movements [38,39]. BA6 plays an important role in working memory and attention [40] and language processing, all of which are disturbed in SCZ and BPD [35].

Our results support that BA6 had more AI in BPD patients, indicating a stronger rightward asymmetry than in SCZ patients. BPD patients had higher GM volume in the right hemisphere for both clusters and less GM volume in the left hemisphere for cluster 3. Other studies support our results that SCZ patients had less GM in the right BA6, while BPD patients had higher volume [7,41]. Motor abnormalities are an overlapping symptom in SCZ and BPD, mainly in the form of neurological soft signs and implicit motor learning disturbance, and the expression of that common pattern of abnormality in these diseases seems to be related to a dysfunction in the cortical-cerebellar-thalamic-cortical circuit [35,39].

The presence of left and right hemisphere disruptions, especially in the supplementary motor cortex in mania, may explain coexisting affective and psychotic symptoms [42]. Morphological abnormalities in the left supplementary motor cortex seem to predispose the development of disturbances of higher motor control during acute psychosis [43]. The hypothesis that the right hemisphere may be dominant in mood regulation correlates with a right hemisphere disturbance in BPD. BPD also has interhemispheric asymmetry of motor cortical excitability, with lower excitability in the right hemisphere, when compared to the contralateral [44], and a reduced regional homogeneity in the left middle frontal gyrus [45].

Higher AI in BPD compared to SCZ was observed in a cluster corresponding to BA37 in the right hemisphere’s inferior and middle temporal parts of the fusiform gyrus, a region essential for language and emotional processing, regulation of responses during face recognition, processing of visual forms of language, and verbal listening stimulus and known to be disturbed in SCZ, contributing to the social and cognitive deficits [46,47,48,49,50].

Knowledge about fusiform gyrus and differences between groups are heterogeneous in the literature, but dysfunction of this area and its circuits are established findings. Previous studies describe that SCZ patients had reduced GM volume and also decreased surface area in the left fusiform gyrus, when compared to BPD patients, suggesting a disturbance in neural functions, affecting the ability to recognize facial emotions and their intensities [14,46,51], besides a lower GM concentration in the bilateral inferior temporal gyrus merging with the fusiform gyrus and the gyrus rectus, either in first episode or chronic SCZ and not only in the left fusiform gyrus [47,48]. Additionally, SCZ patients with higher decreases of GM in the inferior temporal and fusiform gyrus present more severe delusional symptoms, auditory verbal hallucinations, suspicion, and anxiety [46,47,48]. SCZ patients also have ongoing GM reduction in the fusiform gyrus when compared to schizotypal disorders [48]. GM in the anterior fusiform gyrus was found to be smaller in SCZ patients, while the posterior fusiform gyrus had smaller GM volumes in both SCZ patients and schizotypal disorder patients, suggesting that anterior fusiform gyrus modifications and an active pathological process are necessary to develop full-blown SCZ [48,49]. Total fusiform gyrus GM is reduced in SCZ but not in schizotypal disorder, in line with our results [49].

BPD patients had a stronger rightward asymmetry, with higher GM volume in the right hemisphere’s BA37. GM volume in BPD patients is higher than in SCZ patients in bilateral fusiform gyrus [7]. Nonetheless, thickness or volume reduction in the fusiform cortex is associated with increased vulnerability to affective disorders such as BPD, especially in left fusiform gyrus, but in our results, when comparing with SCZ patients, no differences were found in left hemisphere volume in the BPD group, probably because some studies also referred to a similar pattern of cortical thinning in SCZ patients [11,14,15,52,53]. The BPD group also had right-sided intrinsic connectivity distribution in the fusiform gyrus with limbic, prefrontal regions, and cerebellum. Moreover, previous studies also refer to a low local gyrification index in the right fusiform gyrus in BPD patients and decreased functional connectivity to the sensorimotor area and right superior temporal gyrus [50,53]. Although the results are mainly in line with previous studies, some ambiguity is shown in the hemisphere’s GM volumes between conditions.

Finally, our study found that BPD patients had higher asymmetry in the medial frontal gyrus and ACC cluster (corresponding to BA11, BA25, and BA32) in the right hemisphere compared to SCZ patients. In terms of GM volume, BPD patients had higher volume in the right hemisphere, while SCZ patients had higher volume in the left hemisphere.

The orbitofrontal cortex (BA11) plays a role in affective and cognitive processes such as working memory, attentional control, emotional processing, expectation of reward and punishment, decision-making ability, cognitive inhibition, impulse control, and flexibility [54,55,56]. It has a medial subregion—straight gyrus—and a lateral subregion—orbital gyrus; the lateral subregion is involved in the valuation of decision options and is more sensitive to loss of reward, whereas the medial subregion is more sensitive to the presence of rewards [55]. The straight gyrus is an extension of the ACC onto the frontal cortex and has dense inhibitory connections with the auditory cortex and the superior temporal gyrus [19]. Dysfunction in these connections may play a significant role in hallucinations and self-disorder symptoms in SCZ, as bilateral GM reduction of the orbital gyrus, but not of the straight gyrus, has been reported in SCZ patients [55]. BPD patients exhibit GM reductions in both sub-regions of orbitofrontal cortex, while global GM volume in the orbitofrontal cortex is reduced in both SCZ and BPD patients, which is associated with poor executive functions [56]. Reduced GM volume in the orbitofrontal cortex is correlated with positive symptoms in SCZ and deficient emotional processes [54,55], although the specific symptoms related to the orbitofrontal cortex are not clear, given that GM volume decrease has also been correlated with negative symptoms [19].

The ACC (including BA25 and BA32) is responsible for modulating prefrontal and limbic processes, especially in evaluating negative stimuli, attention, acquiring and using social information to guide decisions, and cognitive processes such as inferring others’ emotions [57]. The ACC is divided into two subregions, a ventral, and a dorsal cognitive subregion, with the latter integrating a distributed attentional network and where GM volume reduction in SCZ and BPD has been described [11,13,19,54]. This volumetric finding is more consistent in SCZ and directly relates to the duration of illness [13,19,58].

In line with our results, two other studies showed that decreased GM volume was found in the left ACC in patients with BPD, and that there was higher GM volume in the right ACC in BPD patients compared to the SCZ group [7,59]. ACC has connections with prefrontal, striatal, and limbic regions, including the orbitofrontal cortex, so a disruption in these networks compromises affective regulation and cognition, and this mechanism is described for BPD [13].

In addition, reductions in WM were also found in the right ACC for SCZ patients [12]. GM reduction and hypoactivation of ACC during emotional processing tasks in patients with psychosis are related to negative symptoms, and the greater symptoms such as avolition and anhedonia are, the greater the decrease in ACC activation to pleasant stimuli in SCZ is [9,19,57]. Avolition has a strong relationship with GM asymmetry, and negative symptoms have a strong association with ACC and scales scores of emotional withdrawal and difficulty in abstract thinking [9]. On the other hand, scale scores of stereotyped thinking are more related to the straight gyrus in orbitofrontal cortex [19].

Despite this case-control study’s rigorous design, matching for relevant variables such as gender, age, and inclusion/exclusion criteria, some limitations should be discussed. First, its cross-sectional design as well as a relatively modest sample size, namely of subgroups such as non-psychotic BPD patients, preclude the assessment of some variables, e.g., the influence of psychotic symptoms in BPD. While we assessed current medication use, namely antipsychotics and lithium, its possible effect as a confounder cannot be ruled out.

## 5. Conclusions

In this gray matter asymmetry study, we found brain areas with statistically different asymmetry indexes between SCZ and BPD patients. Namely, we found an asymmetry index higher in BPD patients when compared to SCZ in Brodmann areas 6, 11, and 37 and anterior cingulate cortex and an asymmetry index higher in SCZ patients when compared to individuals with BPD in the cerebellum. While a common critique of psychiatric neuroimaging research has been its relative disconnection from healthcare and patients’ unmet needs, our findings could translate into meaningful gains for clinical practice, besides improving comprehension of specific neurobiological trajectories of SCZ and BPD [60]. While there is not one single area responsible for a specific symptom, a disruption of connectivity, together with structural deficits, such as GM volume asymmetry, might underpin different expressions of these severe mental disorders. These results can, if replicated, be translated into clinical practice to help professionals in the challenging task of differential diagnosis.

## Figures and Tables

**Figure 1 jcm-12-03421-f001:**
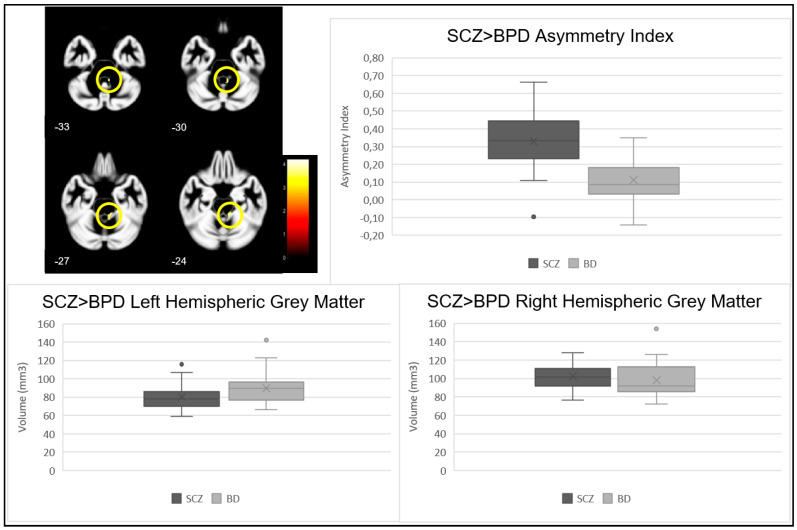
Cluster location (shown in yellow circles), asymmetry index, and right and left GM volumes in clusters showing a significant contrast SCZ > BPD.

**Figure 2 jcm-12-03421-f002:**
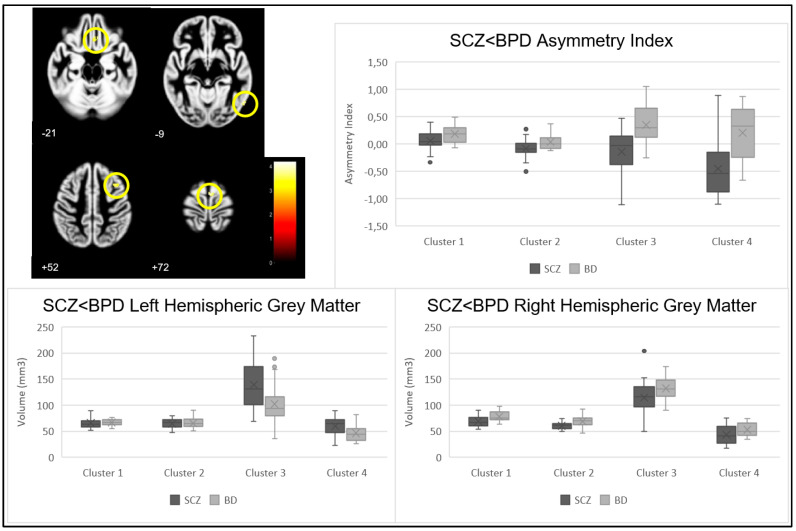
Cluster location (shown in yellow circles), asymmetry index, and right and left grey matter volumes in clusters showing a significant contrast SCZ < BPD.

**Table 1 jcm-12-03421-t001:** Clinical and demographic data of study groups.

	Schizophrenia *n* = 20	Bipolar Disorder *n* = 20	Healthy Controls *n* = 20	Test Statistics	*p*-Value
Gender (Female | Male)	7 | 13	7 | 13	7 | 13	$x^{2}$ 0.000	1.000
Age—years (SD)	31.5 (10.3)	31.65 (10.00)	31.5 (10.3)	F 0.001	0.992
Education—years (SD)	13.6 (3.7)	13.85 (2.64)	14.9 (4.52)	F 0.756	0.474
Age of disease onset—years (SD)	25.6 (6.9)	26.5 (8.8)	-	t −0.276	0.784
Duration of disease—years (SD)	6.0 (7.9)	5.2 (4.3)	-	t 0.297	0.769
Number of lifetime admissions (min–max)	1.25 (0–7)	1.25 (0–4)	-	t 0.000	1.000
Antipsychotic exposure (CPZE)—mg (SD)	380.0 (337.3)	160.8 (272.3)	-	t 2.226	0.032
History of psychotic symptoms	20	16	-	$x^{2}$ 0.035	0.106
History of substance abuse	5	7	-	$x^{2}$ 0.557	0.731
History of suicidal behaviors	4	4	-	$x^{2}$ 0.000	1.000
Psychopathology—BPRS (SD)	35.65 (6.41)	29.11 (2.61)	-	t 3.991	0.000
Functioning—PSP (SD)	80.22 (12.36)	92.00 (4.00)	-	t −3.845	0.001
Insight—ITAQ (SD)	17.12 (3.16)	19.13 (2.22)	-	t −2.100	0.044
Schizo-Bipolar Scale (min–max)	8.00 (7–9)	0.94 (0–2)	-	t 28.356	0.000

BPRS—Brief Psychiatric Rating Scale; CPZE—Chlorpromazine equivalents; ITAQ—Insight and Treatment Attitudes Questionnaires; PSP—Personal and Social Performance Scale; SD—Standard Deviation.

**Table 2 jcm-12-03421-t002:** Clusters’ Description—Significant GM asymmetries between SCZ vs. BPD, SCZ vs. HC, and BPD vs. HC and respective cluster size, MNI coordinates, and corresponding brain region.

	Cluster	*p*-Value	Z Statistic	Cluster Size	MNI Coordinates (X, Y, Z)	Brain Region
SCZ > BPD	1	<0.001	3.78	82	(11, −34, −26)	Right Cerebellum hemisphere, Anterior Lobe, Gyrus Culmen
		<0.001	3.52		(6, −45, −30)	
SCZ < BPD	1	<0.001	3.92	38	(3, −6,72)	Right Cerebrum, Frontal Lobe, Medial Frontal Gyrus, Brodmann area 6
	2	<0.001	3.78	35	(57, −66, −9)	Right Cerebrum, Temporal Lobe, Fusiform, Inferior Temporal, and Middle Temporal Gyrus, Brodmann area 37
	3	<0.001	3.47	60	(33, 14, 52)	Right Cerebrum, Frontal Lobe, Medial Frontal Gyrus, Brodmann area 6
	4	<0.001	3.44	33	(5, 29, −21)	Right Cerebrum, Frontal and Limbic Lobe, Medial Frontal Gyrus and Anterior Cingulate Cortex, Brodmann area 11, 25 and 32
SCZ > HC	1	<0.001	4.37	207	(44, 21, −9)	Right Cerebrum, Frontal Lobe, Inferior Frontal Gyrus, Brodmann area 47
		<0.001	3.82		(39, 23, −17)	
	2	<0.001	3.56	57	(30, −21, 74)	Right Cerebrum, Frontal Lobe, Precentral Gyrus, Brodmann area 4 and 3
SCZ < HC	1	<0.001	3.71	83	(35, −70, 54)	Right Cerebrum, Parietal Lobe, Superior Parietal Lobule, Brodmann area 7
		<0.001	3.59		(29, −64.52)	
	2	<0.001	3.68	33	(17, 11, 10)	Right Cerebrum, Caudate Body
BPD > HC	1	<0.001	4.50	112	(47, 23, 9)	Right Cerebrum, Frontal Lobe, Sub-gyral, White Matter
	2	<0.001	4.43	81	(38, 38, 6)	Right Cerebrum, Frontal Lobe, Sub-gyral, White Matter
	3	<0.001	3.61	33	(15, 11, −3)	Right Cerebrum, Caudate Head
BPD < HC	1	<0.001	4.29	45	(16, −60, 60)	Right Cerebrum, Parietal Lobe, Precuneus, Brodmann area 7

**Table 3 jcm-12-03421-t003:** ANCOVA between SCZ and BPD patients’ left and right grey matter volumes in significant clusters with total intracranial volume as a covariate.

		*p*-Value	
	Cluster	Left Hemisphere	Right Hemisphere
SCZ > BPD	1	0.080	0.406
SCZ < BPD	1	0.731	0.007
	2	0.900	0.002
	3	0.012	0.078
	4	0.018	0.044

## Data Availability

The data presented in this study are available on request from the corresponding author.

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
