# Peer review of "Brain Hemispheric Asymmetry in Schizophrenia and Bipolar Disorder"

_jcm, 2023, doi:10.3390/jcm12103421_

Round 1
Reviewer 1 Report
The manuscript by Pinto et al., entitled "Brain hemispheric asymmetry in schizophrenia and bipolar disorder" and found to be a valuable contribution to the field of neuropsychiatry. The study aimed to investigate the brain hemispheric asymmetry in patients with schizophrenia (SCZ), bipolar disorder (BPD), and healthy controls and evaluate whether these asymmetry patterns could differentiate and establish boundaries between these two partially overlapping severe mental disorders.
The authors used a fully automated voxel-based morphometry (VBM) approach to assess the structural brain hemispheric asymmetry in magnetic resonance imaging (MRI) anatomical scans in 60 participants (SCZ=20; BPD=20; healthy controls = 20), all right-handed and matched for gender, age, and education. The results showed that there were significant differences in grey matter asymmetry between patients with SCZ and BPD, between SCZ patients and healthy controls (HC), and between BPD patients and HC. The authors found a higher asymmetry index (AI) in BPD patients when compared to SCZ in Brodmann areas 6, 11, and 37 and anterior cingulate cortex, and an AI higher in SCZ patients when compared to BPD in the cerebellum.
The authors' conclusion that their study found significant differences in brain asymmetry between patients with SCZ and BPD is well-supported by their results, which could be translated to clinical practice to help clinicians in the differential diagnosis. The conclusion is well-supported by the results and the implications of the findings are significant. The findings are further statistically significant as they provide a foundation for exploring MRI-detected structural brain changes as potential biological markers for differential diagnosis and helping to understand disease-specific abnormalities. The study is well-designed, and the methods are adequately described.
However, there are a few minor concerns that the authors should address. First, the authors should discuss the potential implications of the study's findings for treatment or prognosis. Second, while the results are promising, the study's sample size is relatively small, and future research should aim to replicate the findings in a larger sample. Finally, the authors should include a statement on the limitations of their study, such as the cross-sectional design and the use of a single imaging modality.
Author Response
We thank the reviewers for the constructive reviews and valuable suggestions provided. We have addressed all the raised points in a comprehensive manuscript revision, and the modifications have significantly improved it. Here, we respond to each issue and indicate corresponding manuscript modifications, highlighted in the submitted marked-up version.
Reviewer #1
The manuscript by Pinto et al., entitled "Brain hemispheric asymmetry in schizophrenia and bipolar disorder" and found to be a valuable contribution to the field of neuropsychiatry. The study aimed to investigate the hemispheric brain asymmetry in patients with schizophrenia (SCZ), bipolar disorder (BPD), and healthy controls and evaluate whether these asymmetry patterns could differentiate and establish boundaries between these two partially overlapping severe mental disorders.
The authors used a fully automated voxel-based morphometry (VBM) approach to assess the structural brain hemispheric asymmetry in magnetic resonance imaging (MRI) anatomical scans in 60 participants (SCZ=20; BPD=20; healthy controls = 20), all right-handed and matched for gender, age, and education. The results showed significant differences in grey matter asymmetry between patients with SCZ and BPD, between SCZ patients and healthy controls (HC), and between BPD patients and HC. The authors found a higher asymmetry index (AI) in BPD patients when compared to SCZ in Brodmann areas 6, 11, and 37 and anterior cingulate cortex, and an AI higher in SCZ patients when compared to BPD in the cerebellum.
The authors' conclusion that their study found significant differences in brain asymmetry between patients with SCZ and BPD is well-supported by their results, which could be translated to clinical practice to help clinicians in the differential diagnosis. The conclusion is well-supported by the results and the implications of the findings are significant. The findings are further statistically significant as they provide a foundation for exploring MRI-detected structural brain changes as potential biological markers for differential diagnosis and helping to understand disease-specific abnormalities. The study is well-designed, and the methods are adequately described.
However, there are a few minor concerns that the authors should address. First, the authors should discuss the potential implications of the study's findings for treatment or prognosis. Second, while the results are promising, the study's sample size is relatively small, and future research should aim to replicate the findings in a larger sample. Finally, the authors should include a statement on the limitations of their study, such as the cross-sectional design and the use of a single imaging modality.
R1. We thank the reviewer for the many positive comments and insightful comments.
We rephrased the discussion of the study’s limitations in accordance – page 10.
“Despite this case-control study's rigorous design, matching for relevant variables such as gender and age and namely inclusion/exclusion criteria, some limitations should be discussed. First, its cross-sectional design, as well as a relatively modest sample size, namely of subgroups such as non-psychotic BPD patients, precludes the assessment of some variables, e.g., the influence of psychotic symptoms in BPD. While we assessed and controlled for current medication use, namely antipsychotics, and lithium, its possible effect as a confounder cannot be entirely ruled out.
Reviewer 2 Report
The authors wanted to explore the potential grey matter assymetry that could be observed in BPD and SCZ
I think the study is well done, however i have a few comments.
1- intracranial volume, age, sex are important factors that need to be added as controlling variables in such analysis ( if the relevance of age and sex could always be discussed, TIV is a standard and mandatory factor)
2- similarly, for between patients comparison, medications should also be accounted for.
3- finally, the discussion is extremely long and hard to follow.
It needs to be shortened to make it clearer.
Author Response
We thank the reviewers for the constructive reviews and valuable suggestions provided. We have addressed all the raised points in a comprehensive manuscript revision, and the modifications have significantly improved it. Here, we respond to each issue and indicate corresponding manuscript modifications, highlighted in the submitted marked-up version.
Reviewer #2
The authors wanted to explore the potential grey matter assymetry that could be observed in BPD and SCZ
I think the study is well done, however i have a few comments.
1- intracranial volume, age, sex are important factors that need to be added as controlling variables in such analysis (if the relevance of age and sex could always be discussed, TIV is a standard and mandatory factor)
R1. We thank the reviewer for pointing out this important issue.
Indeed, the inclusion of age and sex can be discussed. However, the groups in our study are perfectly sex- and age-matched; thus, these factors would hardly affect between-groups differences.
On the other hand, the reviewer is correct in stating that TIV is a standard covariate in the VBM analysis of GM. Groups are also matched on TIV, as reported in our previous study with the same participants.
> https://www.sciencedirect.com/science/article/pii/S2213158220300577#fig0001
Nonetheless, we acknowledge the reviewer’s comment and have re-done the analysis of testing between-groups differences with ANCOVA using TIV as a covariate. The results are maintained, and only a few p-values were slightly different. We have updated the analysis description and all results (statistics and p-values) in the manuscript.
2- similarly, for between patients comparison, medications should also be accounted for.
R2. We thank the reviewer for the comment.
When discussing the limitations of this work, we report the possible role of medication – page 10.
“While we assessed for current medication use, namely antipsychotics, and lithium, its possible effect as a confounder cannot be ruled out.”
Given that SCZ patients, as a class, had higher antipsychotic exposure (CPZE) (mean: SCZ=380.0mg; BPD = 160.8mg), we chose not to include it as covariable, given its predictable collinearity with the group factor.
3- finally, the discussion is extremely long and hard to follow.
It needs to be shortened to make it clearer.
R3. We thank the reviewer for this relevant improvement suggestion. We revised the discussion to make it clearer and easier to follow. We cut over 750 words and improved the text flow.
Round 2
Reviewer 2 Report
The authors addressed all my comments.